# Integrated quantum optical phase sensor in thin film lithium niobate

Hubert S. Stokowski [1], Timothy P. McKenna[2], Taewon Park[1,3], Alexander Y. Hwang[1], Devin J. Dean [1], Oguz Tolga Celik[1,3], Vahid Ansari [1], Martin M. Fejer[1] & Amir H. Safavi-Naeini [1] ✉

The quantum noise of light, attributed to the random arrival time of photons from a coherent light source, fundamentally limits optical phase sensors. An engineered source of squeezed states suppresses this noise and allows phase detection sensitivity beyond the quantum noise limit (QNL). We need ways to use quantum light within deployable quantum sensors. Here we present a photonic integrated circuit in thin-film lithium niobate that meets these requirements. We use the second-order nonlinearity to produce a squeezed state at the same frequency as the pump light and realize circuit control and sensing with electro-optics. Using 26.2 milliwatts of optical power, we measure $(2.7 \pm 0.2)\%$ squeezing and apply it to increase the signal-to-noise ratio of phase measurement. We anticipate that photonic systems like this, which operate with low power and integrate all of the needed functionality on a single die, will open new opportunities for quantum optical sensing.

Squeezed states of light exhibit fluctuations in their quadrature amplitude below that of the quantum vacuum[1]. This property makes them indispensable resources for increasing the sensitivity of optical measurements. In interferometry with classical light, signal-to-noise-ratio scales as $\sqrt{N}$ with the total number of detected photons $N$. As a result, the imprecision in determining the optical phase, or phase sensitivity, follows $1/\sqrt{N}$, referred to as the quantum noise limit (QNL). In this case, the signal-to-noise-ratio (SNR) is limited by photon shot noise. To increase the SNR within the confines of the QNL, we must increase $N$—by using more optical power or extending the measurement time. These approaches are often undesirable or impossible due to technical or fundamental limitations. Intriguingly, the QNL can be overcome by injecting squeezed states of light into the dark port of the interferometer[2]. Ultimately, this approach may lead to a truly quantum-limited Heisenberg scaling $1/N$ of sensitivity[2–4]. Over the past forty years, impressive efforts have been put into designing squeezed light sources and integrating them into gravitational wave detectors. These detectors now routinely obtain quantum-enhanced sensitivity exceeding 2 dB[5,6]. Similar ideas were also extended into other laboratory demonstrations, e.g., in distributed phase sensing[7] and enhanced signal-to-noise-ratio in biological sensing[8] and microscopy[9].

We aim to demonstrate an integrated quantum optical sensor that generates and uses squeezed light to perform measurements beyond the QNL. Our sensor moves much of the functionality often implemented on the optical table—e.g., generating pump light, locking phases, and implementing an interferometer, onto the chip itself. We use thin-film lithium niobate (LN)[10] to achieve all of the functions in a monolithic circuit. In the last few years, integrated thin-film LN devices have demonstrated efficient light generation covering wavelengths from mid-infrared to blue[11–14], optical parametric oscillators[15,16], dispersion engineering[17–19], EO modulation, tuning[20,21], and frequency combs[22]. This article combines nearly all of the aforementioned capabilities in a single LN chip and demonstrates an integrated quantum sensor. We leverage the strongest electro-optic coefficient of LN ($r_{33} = 31$ pm/V)[23] to actively control the power and phase of light within a photonic integrated circuit (PIC) and interface it with periodically poled LN (PPLN) waveguides that support efficient nonlinear processes. The strong second-order optical nonlinear coefficient $\chi^{(2)}$, accompanied by quasi-phase-matching in PPLN, allows us to achieve efficient second harmonic generation (SHG) and squeezed light generation through optical parametric amplification (OPA).

[1]Department of Applied Physics and Ginzton Laboratory, Stanford University, Stanford, CA 94305, USA. [2]Physics & Informatics Laboratories, NTT Research, Inc., Sunnyvale, CA 94085, USA. [3]Department of Electrical Engineering, Stanford University, Stanford, CA 94305, USA. ✉e-mail: safavi@stanford.edu

In the degenerate parametric amplification process, photons are generated in pairs with a fixed phase relationship through the interaction $\hat{H}_I \propto (\hat{a}^2 + \hat{a}^{\dagger 2})$. The resulting state of the light is squeezed. Following the first demonstration of a squeezed light source by parametric down-conversion (PDC) in an optical cavity[24], numerous other approaches, including single-pass PDC[25,26], semiconductor lasers, optical fibers, atomic ensembles[1], and ponderomotive squeezing with mechanical oscillators[27] have successfully demonstrated varied amounts of squeezed light. Optical materials with intrinsic second- or third-order nonlinearities may be the most versatile integrated sensors approach. Recent demonstrations of integrated photonic circuits in silicon nitride[28–30] proton-exchanged LN[31], and thin-film LN[32,33] have shown that low power and scalable sources of quantum light are possible.

Here, we focus on integrating these developments into a complex photonic circuit that can be used as a cost-effective and self-contained sensor. Because of this goal, we limit the optical power to the typical range of a commercial diode laser. Our goal, like those of the first integrated analog electronic circuits, is to demonstrate a path towards a deployable technology as opposed to realizing record-breaking performance. From a photonic system perspective, the squeezed source is only one part of a larger optical sensor. Combining emerging squeezed light sources with more complex PICs is a promising avenue for developing deployable optical sensors with quantum-enhanced sensitivity. So far, achieving sub-shot-noise sensitivity in optical metrology has required complex setups with multiple modulators, lasers, and optical cavities. This has limited the domain of quantum optical sensing to experiments with high complexity[5,6,9] that can address only a few of the possible application spaces—even for experiments that use new integrated quantum light sources. Our work combines emerging squeezed light sources with a complex PIC to realize a nearly complete on-chip quantum sensor, thus opening a promising avenue for deployable optical sensors with quantum-enhanced sensitivity.

## Results

In this work, we use the X-cut thin film of lithium niobate on an insulator (LNOI) platform to build a sensor with an integrated source of squeezed light for quantum-enhanced measurement. Figure 1a outlines a scheme of the PIC implemented in our chip with three major components—input tunable beam splitter, squeezer, and signal interferometer. We pattern eight such PICs in a single chip, which we show on a microscope picture in Fig. 1b. We design the squeezer to generate a sub-shot noise state of light at the same frequency as the pump light, enabling interference with the local oscillator (LO) extracted from the original beam. Active EO circuitry controls interferometers and the LO phase, which enables using the system for both squeezing characterization and quantum measurement. Using lensed SMF-28 fiber, we interface our device with an off-the-shelf, ultra-low noise DBR laser and operate the system in CW mode at 1544 nm. The output ports of the beam splitter out-couple light into two lensed multimode fibers, which we connect to a balanced photoreceiver off-chip. Both the laser[34,35] and detectors[36] can be further integrated with the PIC to increase the system efficiency and improve the squeezing visibility.

### Device design

The squeezer subsystem comprises two PPLN waveguides and a series of directional couplers for spectral filtering. We design the waveguide geometry to maximize the nonlinear normalized efficiency (see Methods for details). Fundamental harmonic (FH) light first enters a

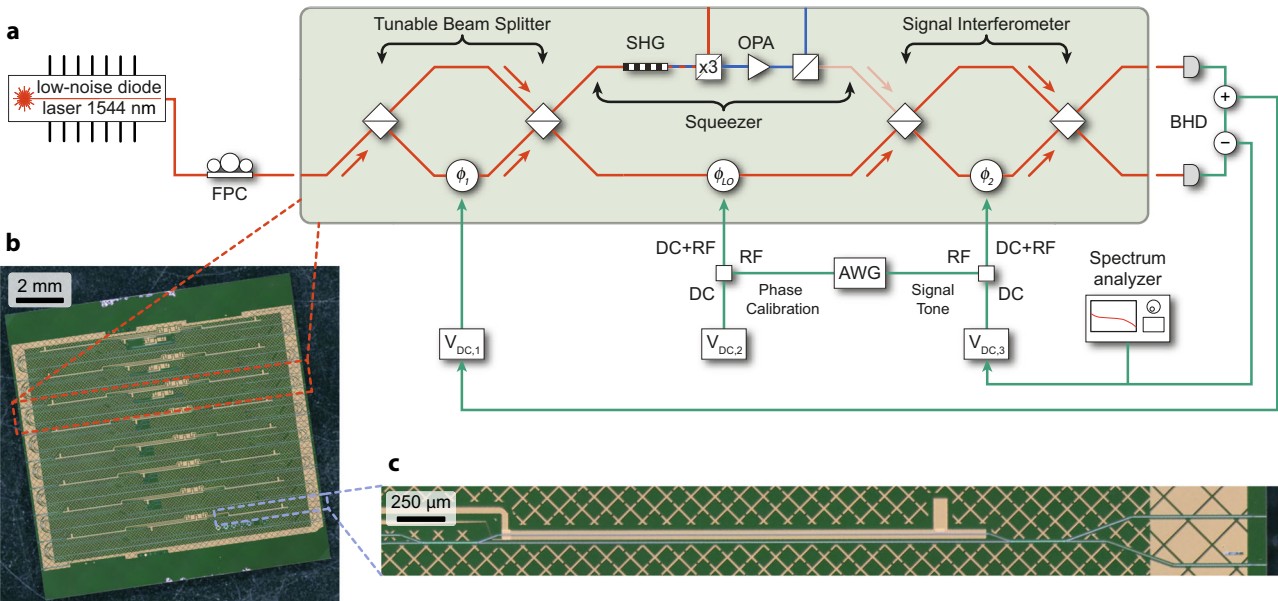

**Fig. 1 | Squeezing and quantum-enhanced measurement PIC. a** Schematic of a photonic integrated circuit for squeezing and quantum-enhanced measurement. Light from an external, integrated laser is polarization-controlled and coupled through a cleaved facet. The PIC splits light into two paths at the tunable beam splitter: the squeezer pump and the local oscillator. The two beams are recombined at the signal interferometer to perform a quantum-enhanced measurement of a small RF signal. The signal interferometer also serves as a 50/50 beam splitter for squeezing characterization in BHD. Light is detected off-chip with a balanced photoreceiver connected to the chip with two lensed multimode fibers. Sum and difference signals provide feedback for locking output splitting ratio and LO power level. The setup contains an additional variable optical attenuator (VOA) just after the laser in the shot noise calibration measurements but is omitted for the squeezing measurements to reduce insertion loss. **b** Microscope picture of a chip with eight PICs, the red rectangle highlights a single device, blue outlines a single signal interferometer. A large-scale gold cross-pattern covers the entire chip to prevent stray light from interfering with guided modes. **c** Microscope picture of an MZI serving as a signal interferometer. The asymmetric design provides phase modulation to one of the arms of the interferometer through gold electrodes routed to an RF probe. The light blue lines in **b** and **c** correspond to optical waveguides; gold structures are elements of the EO-tuning circuitry. FPC fiber polarization controller, SHG second harmonic generation, OPA optical parametric amplification, BHD balanced homodyne detection, AWG arbitrary waveform generator.

PPLN waveguide designed to phase match a second harmonic generation process using periodic poling with a period of around 3.7 µm. Tight mode confinement and strong second-order optical nonlinear coefficient of LNOI yield highly efficient SHG. Next, we filter out the residual FH by passing the light through three directional couplers. By design, each directional coupler transfers the majority of FH while keeping the second harmonic (SH) light in the original waveguide; hence the structure acts as a dichroic beam splitter. The SH light then loops back to another PPLN waveguide, realizing a phase-sensitive OPA of the quadrature fluctuations at the FH frequency, squeezing one quadrature and anti-squeezing the other. The amount of squeezing generated in this structure is given by:

$$\frac{\langle \delta \hat{Y}^2 \rangle}{\langle \delta \hat{Y}_{vac}^2 \rangle} = \exp\left(-2L\sqrt{\eta P_{in}} \tanh\left(L\sqrt{\eta P_{in}}\right)\right), \quad (1)$$

where $\langle \delta \hat{Y}^2 \rangle$ is the variance of the quadrature squeezed with respect to vacuum $\langle \delta \hat{Y}_{vac}^2 \rangle$, $L$ is the length of the poled waveguides, $\eta$ is the normalized nonlinear conversion efficiency[11], and $P_{in}$ is the telecom pump power (see Methods for details). Finally, the SH light is filtered out using the same directional coupler design, and the squeezed state propagates to the signal interferometer at the output of the PIC. We route the light filtered out after the SHG and OPA sections to the chip facet and use them as monitor ports.

Active EO circuitry controls optical power splitting and the LO phase within the PIC. The tunable beam splitter at the input and the signal interferometer at the output are both intensity modulators. Both consist of a Mach–Zehnder interferometer (MZI) with a phase modulator in one arm. The former is typically DC-biased at phase $\phi_1$ either close to zero, to direct most of the input light into the squeezer, or set to $\pi$, to send most of the light into the LO and characterize shot noise. The latter MZI, shown in Fig. 1c, is always locked with a DC bias to make the power at the two outputs equal for a balanced homodyne detection (BHD) measurement. We use the same phase modulator design for quadrature selection in the squeezer path through $\phi_{LO}$ control. In addition to DC bias, we apply RF tones to the LO phase modulator and the signal interferometer. The RF phase modulation on the LO serves as voltage-to-phase calibration and quantifies the light leakage in the squeezer $\sqrt{\varepsilon} = \alpha_{Leakage}/\alpha_{LO}$, while the RF phase modulation imposed on the signal interferometer forms the signal we measure. The mentioned leakage occurs only when a significant amount of power pumps the squeezer subsystem. There is no leakage during the shot noise calibration since there is no light in the squeezing path.

## Subsystem characterization

We calibrate the shot noise level by routing all the optical power to the local oscillator path. For this purpose, we bias the tunable beam splitter at the input with a DC voltage bias of 47 V. In this configuration, we use the local oscillator to probe the vacuum state sent to the opposing port of the signal interferometer. For the BHD measurement, we lock the signal interferometer to a splitting ratio of 50:50. We achieve this by controlling the difference between the intensity of two outputs and locking its DC part to zero. Next, we step the power of the input laser using an external variable optical attenuator (VOA), such that the power in the local oscillator waveguide varies between 0 and 8.8 mW and record the RF spectra of the balanced homodyne detector, as shown in Fig. 2a. Finally, we integrate the noise power and plot it against the LO power in Fig. 2b, observing a linear dependence as expected from a shot noise-limited measurement. Note that the linearity of our measurement implies that there is no excess intensity noise from the laser. Hence, we measure only shot noise and can use it as a reference for further squeezing measurements. The mean squared error of the linear fit defines our uncertainty of the shot noise

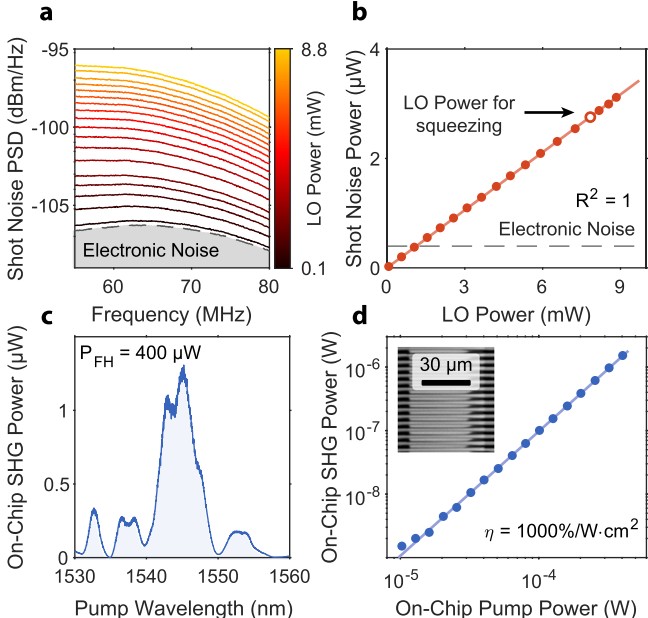

**Fig. 2 | PIC subsystems characterization. a** Shot-noise power spectral density for different LO powers on the chip. We characterize our setup to find LO power yielding signals about 10 dB above the electronic noise floor. **b** Integrated noise from Fig. 2a vs. LO power follows a perfect linear trend, proving that the system is limited by shot noise at the target LO power of 7.84 mW. **c** Example SHG phase-matching curve, deviation from ideal $sinc^2$ shape results from small geometry variation along the waveguide length. **d** Measured on-chip SHG peak power vs. FH pump power. Linear fit yields normalized efficiency of the nonlinear process of about 1000%/(W·cm²). The inset shows an SHG microscope image of high-quality periodic poling of the thin-film lithium niobate. We pattern waveguides in the area where poling extends through the entire thickness of the LN film.

measurement. Based on the shot noise calibration, we choose LO power for the squeezing experiments at around 7.84 mW, where the optical shot noise dominates our detection. We highlight this operating point in Fig. 2b. Note that the noise and LO power are recorded simultaneously, and the detector noise determines the uncertainty of the LO power below a relative deviation of $3 \times 10^{-5}$.

The squeezer relies on the performance of the periodically poled waveguides in the SHG and OPA sections. We characterize our periodic poling by measuring the quasi-phase-matching transfer function of a diagnostic waveguide patterned adjacent to our SHG and OPA segments such that it passes through the same poling region (see Methods for measurement details). An example QPM transfer function is shown in Fig. 2c. This measurement gives us the SHG response decoupled from the rest of the PIC components. Because the SHG and OPA take advantage of the same nonlinearity and poled waveguides are identical, we can directly use the nonlinear efficiency obtained in the SHG measurements to benchmark the OPA performance. Note that the diagnostic waveguide is 8.1 mm long, whereas the one in the primary device is 10 mm long. Given the significant length of the fabricated waveguide, nonuniformities distort the ideal $sinc^2$ shape of the nonlinear transfer function[37] and reduce the peak nonlinear efficiency. We measure the intensity of generated SH light as a function of pump power controlled by external VOA. In Fig. 2d, we plot data and linear fit to extract a maximum normalized efficiency of about 1000%/(W·cm²), lower than the simulated 4000%/(W·cm²) for an ideal poled structure. We use this number further to estimate the amount of parametric amplification and the expected squeezing level. Because of the nonuniformities, the quadratic scaling of the efficiency with length is not exact. However, we neglect this because the length difference between the test and device waveguides is small. The inset of Fig. 2d shows a

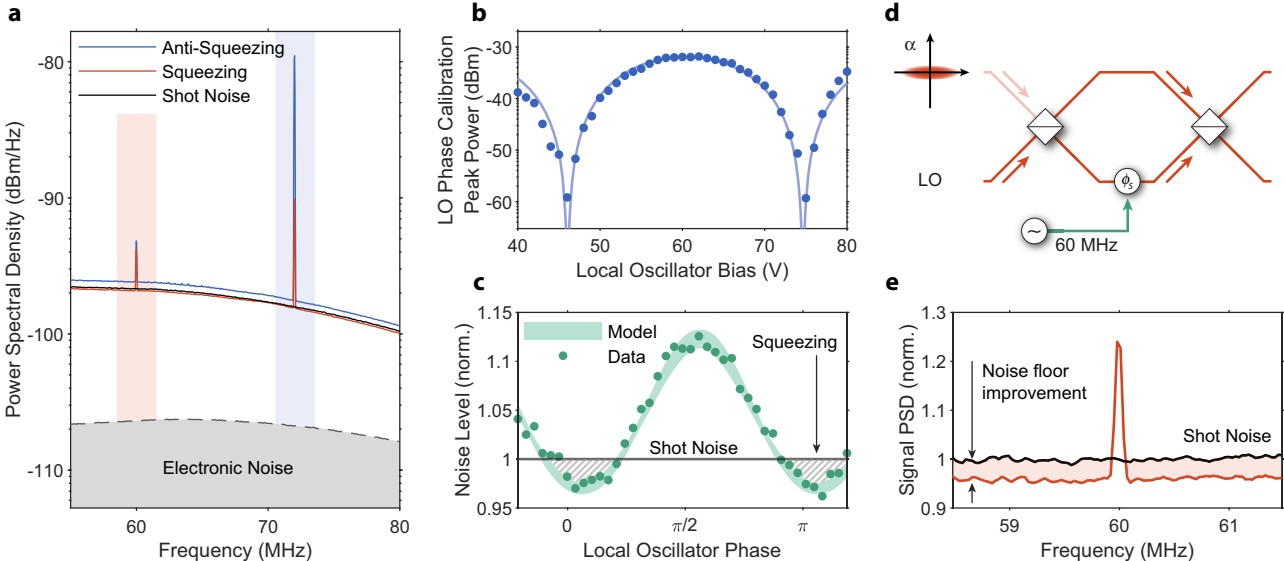

**Fig. 3 | Squeezing and quantum-enhanced detection. a** Power spectral density for maximum squeezing (orange line) and maximum anti-squeezing (blue line), collected while pumping the squeezer with 18.4 mW of FH. The black line provides a shot noise measurement reference. We highlight two RF peaks resulting from modulation of the LO phase shifter (blue highlighted area) and signal interferometer (orange highlighted area). **b** Local oscillator calibration peak power vs. DC bias voltage of the phase shifter. The solid line indicates the fit line, from which we find $V_\pi = 28.6$ V and light leakage into the squeezing path $\varepsilon \approx 4.0\%$. **c** Integrated noise power as a function of the LO phase (calibrated in Fig. 3b). The gray-dashed area represents the characteristic sub-shot noise behavior of a squeezed state. Misalignment of the squeezed state and LO phases is caused by the path length

difference between the two paths. The green highlighted area represents the model, including squeezer leakage, detection efficiency, and normalized efficiency. The uncertainty of the normalized efficiency and phase sets the model bounds. **d** Quantum-enhanced measurement scheme—squeezed state feeds into a dark port of the MZI along with the LO entering the bright port. Phase modulation applied in one of the arms can be detected with enhanced SNR compared to a classical measurement. **e** Measurement of a small RF tone with noise floor suppressed with squeezing. We align the LO phase with maximum squeezing by DC tuning and measure the RF power spectrum through a BHD. For a −83 dBm modulation, we observe about $(3.7 \pm 1.0)\%$ improvement of the SNR compared to a shot-noise-limited measurement.

second harmonic microscope image of periodic poling used in the fabrication process. Black stripes on the sides of the picture are metal finger electrodes, and light gray shapes stretching between them correspond to inverted crystal domains.

## Squeezing and quantum-enhanced sensing

We characterize squeezing by directing most of the laser light into the squeezer and holding the LO power constant. We drive the squeezer subsystem with 18.4 mW of optical power. Given the calibrated normalized efficiency, we estimate SH power at around 3.4 mW to generate about $(1.5 \pm 0.1)$ dB of squeezing on the chip. We perform a BHD measurement to verify this by beating the squeezed state with the local oscillator at the signal interferometer. We lock the phase shift within the output interferometer such that the DC part of the difference signal is zero. At the same time, we lock the input beam splitter to keep the sum of photodiode readouts at a constant value, ensuring the LO power is fixed at around $(7.84 \pm 0.03)$ mW. We follow this locking scheme to work around the slow phase drift observed in EO devices[38,39]. A simple procedure that periodically updates the voltage at the input beam splitter and signal interferometer at a low frequency (around 1 Hz) is sufficient. Finally, we measure the power spectral density, as shown in Fig. 3a, and reference it to the shot noise measurement (black line). We perform this measurement as a function of the LO phase, controlled through a DC voltage applied to a phase shifter in the LO path. Blue and orange traces in Fig. 3a correspond to the maximum anti-squeezing and maximum squeezing, respectively. In addition to probing broadband noise, we introduce two RF signals corresponding to peaks at 72 and 60 MHz. The former originates from an RF signal applied to the LO phase shifter, which we highlight in blue, and the latter corresponds to a tone on the signal interferometer, which we highlight in orange.

We verify the phase-sensitive nature of the generated squeezed light by stepping the LO phase shifter bias. Due to a small power leakage of the FH in the squeezer path, $\varepsilon$, we can observe the RF peak introduced at the LO phase shifter at 72 MHz. This signal is proportional to $\varepsilon \sin^2(V_{DC}/V_\pi \cdot \pi)$ (see Methods for details). By fitting the peak power as a function of LO bias voltage in Fig. 3b, we establish the power leakage ratio $\varepsilon$ at around $(4.0 \pm 1.0)\%$, which we use to estimate the filter extinction ratio at around 18 dB (see Methods for details on filter design and characterization). This amount of leakage enables us to find the relative phase between the LO and squeezed state. Note that a single phase shifter on the LO allows us to set any relative phase between the squeezed light and the LO. Consequently, we can characterize the squeezed state or align the phases to only sample the sub-shot noise quadrature in the quantum-enhanced measurement. The observed amount of leakage does not significantly impact the squeezing visibility in our system (see Methods for details). Using the fit, we also find the half-wave voltage of our LO to be $28.6 \pm 0.3$ V and convert the DC bias to phase. The half-wave voltage agrees with our design (see Methods for details). In the squeezing measurement, in Fig. 3c, we plot the integrated noise floor over the measurement bandwidth vs. calibrated LO phase, excluding the two RF peaks and a small, broad feature between 67 and 72 MHz. Plotted noise is normalized to shot noise, in which uncertainty is smaller than the width of the line. Moreover, we continuously monitor the stability of the DC optical power readout on both detectors to ensure that the LO power is constant. The relative deviation of the LO power during the squeezing measurement is below 0.3%, which allows us to detect changes to the noise on the order of several percent (see Methods for details). We see expected periodic behavior, where the minimum and maximum correspond to probing squeezed and anti-squeezed field quadratures, respectively. Instead of performing fast phase sweeps, we maintain the LO phase at each tuning point for roughly 13 s and obtain

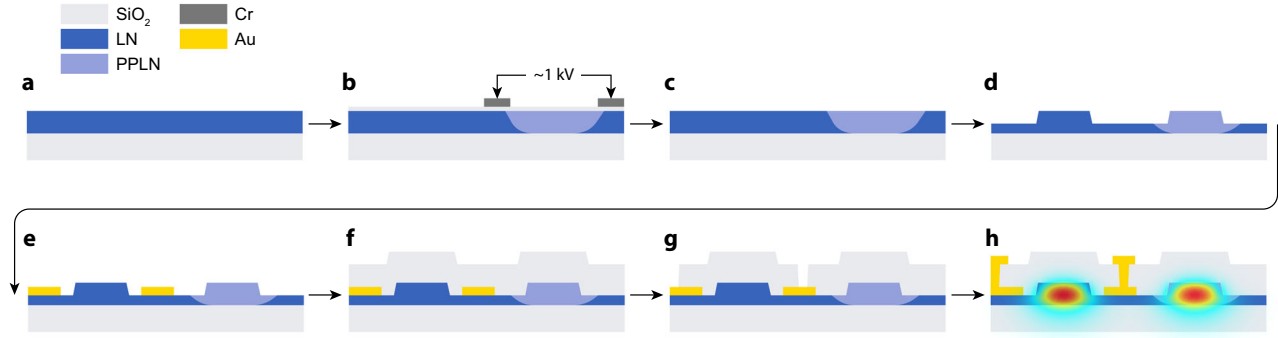

**Fig. 4 | Device fabrication procedure. a** We start with a 500 nm thin-film LN on an insulator. **b** For periodic poling of the LN film, we use chromium electrodes deposited on a 100 nm silica insulation layer. Next, we apply a high-voltage pulse to invert crystal domains. **c** Chromium electrodes and silica layer are removed after poling using chromium etchant and buffered oxide etchant. **d** Waveguides are patterned using electron beam lithography and argon ion milling. **e** We pattern a bottom layer of gold directly on LN to increase the electric field within the wave-guide region. **f** Wavegues are covered with around 700 nm of silica. **g** We open vias in the cladding to provide electrical contact to the buried electrodes. **h** We pattern the top layer of gold to connect our devices to the external probe.

the full RF spectrum. This highlights the stability of our system, a necessary ingredient for a stable and metrologically useful source of squeezed light. We measure $(2.7 \pm 0.2 \, (\text{stat}) \pm 0.3 \, (\text{syst}))\%$ of squeezing and $(12.3 \pm 0.2 \, (\text{stat}) \pm 0.3 \, (\text{syst}))\%$ of anti-squeezing.

Limited squeezing visibility and squeezing/anti-squeezing asymmetry result from the detection chain efficiency of 20% (see Methods for more details) and squeezer leakage $\varepsilon \approx 4\%$. Accounting for these factors, we expect the level of squeezing on-chip to be around $(1.5 \pm 0.1)$ dB, which agrees with our estimation based on the normalized efficiency. The detection chain loss consists mainly of the fiber-chip coupling with contributions from the small waveguide propagation loss and non-ideal quantum efficiency of the photodiodes. The fiber-chip coupling is also a limiting factor in the available optical pump power. Total detection chain efficiency can improve by up to $\approx 80\%$ by reducing optical loss. This improvement will increase the amount of squeezing generated on the chip (due to higher available optical power) and squeezing visibility (due to suppressed loss of detection). Implementing straightforward PIC layout improvements to reduce the propagation loss, improved fiber-to-chip coupling[40–42], and higher quantum efficiency photodiodes can result in an achievable 4.5 dB squeezing or SNR improvement by 280% using the same off-the-shelf integrated laser. See the Methods section for more details on the loss in the current device and the path toward loss reduction.

Finally, we use the signal interferometer to perform a quantum-enhanced measurement of an RF signal. The classical approach to increasing SNR is by using more optical power. However, this strategy is limited −increasing optical power in electro-optic-modulators causes a series of parasitic effects related to the second- and third-order nonlinearity. Especially the photorefractive effect can cause instability of electro-optic modulators. Moreover, photodetectors suffer from saturation and nonlinearity, and can be damaged with large optical power. Using the additional power to generate squeezing offers an alternative. In this case, the chip configuration is the same as in the squeezing measurement; most light pumps the squeezer, and the LO power is fixed at $(7.84 \pm 0.03)$ mW. We use the signal interferometer to perform a quantum-enhanced measurement by injecting squeezed state into its dark port, as shown in Fig. 3d. Based on our squeezing calibration, we set the LO phase to align with the maximum observed squeezing and perform the measurement. For that purpose, we apply a small RF tone to the output interferometer electrode. We set the RF amplitude to −83 dBm, corresponding to the root-mean-square voltage of 15.8 µV at 60 MHz. This signal can be detected in the BHD measurement, as shown in Fig. 3e, and it is not sensitive to the LO phase, which is set before the light enters the output interferometer. As a result, the measurement noise floor is reduced with respect to the

shot noise and results in about $(3.7 \pm 1.0)\%$ SNR improvement. This data was taken at a different time than the LO phase sweep dataset shown in Fig. 3c, and so we attribute the slightly higher level of observed squeezing to slightly different experimental conditions. The uncertainty on the SNR improvement is also more significant than the squeezing measurement since it comes from just one dataset, as opposed to the entire LO phase sweep.

## Discussion

Complex quantum photonic integrated circuits, like the one presented in this work, extend the reach of quantum technologies into new domains. The integration allows us to generate quantum light and leverage its properties within the same circuit, enabling more stable and lower power operation. These critical advantages point toward new classes of mobile and deployable sensors that are natively "quantum-compatible". In the work we present here, we benchmark a sensor by detecting voltage signals imparted as phase upon an optical field. Many other interferometric measurements are today limited by shot noise. These measurements, including refractive index sensing, optomechanical force, acceleration[43], and mass sensing[44], will directly benefit from integrating the quantum apparatus, as demonstrated here. Moreover, squeezing and squeezed light are crucial elements of numerous quantum sensing, communications[45], and computing protocols[46,47]. Quantum PICs, such as those shown here, are promising and scalable approaches for building these systems.

## Methods
### Fabrication
We fabricate our device with X-cut thin-film LNOI (lithium niobate on insulator, NanoLN), following the procedure outlined in Fig. 4. The material stack consists of 500 nm of lithium niobate bonded to a 2 µm-thick silica layer on top of a silicon handle wafer, as shown in Fig. 4a. Fabrication starts with poling the thin-film. In the first step (Fig. 4b), we pattern Cr electrodes on an insulating layer of 100 nm SiO$_2$ using electron beam lithography (JEOL 6300-FS, 100-kV) and the liftoff process. The poling period is around 3.7 µm, and we design it for phasematching between our waveguide's 1550 and 775 nm modes. Next, we apply high-voltage pulses to flip crystal domains[16,48]. After poling, we remove the electrodes with chromium etchant and buffered oxide etchant (Fig. 4b).

The photonic circuit is patterned with a FOX-16 mask and electron beam lithography and transferred to the LN layer with an argon ion mill (Fig. 4d). The etch depth is 300 nm, and the waveguide width is 1.2 µm.

Metal electrodes for EO tuning are patterned with the same process as the poling electrodes but made out of 100 nm gold (Fig. 4e).

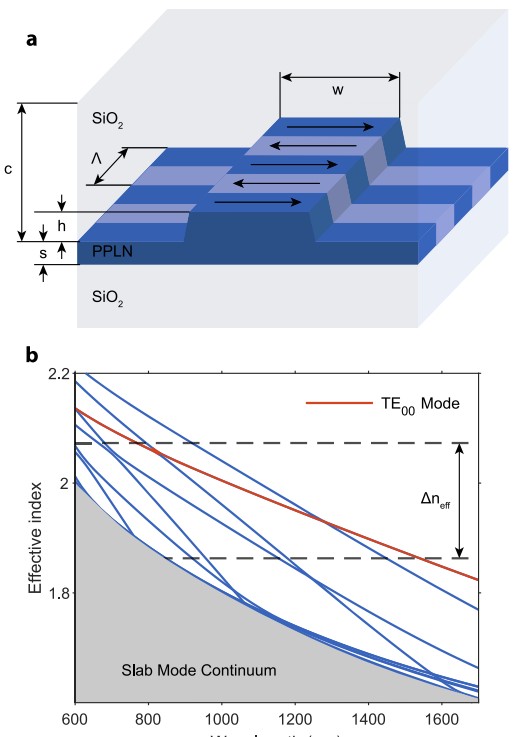

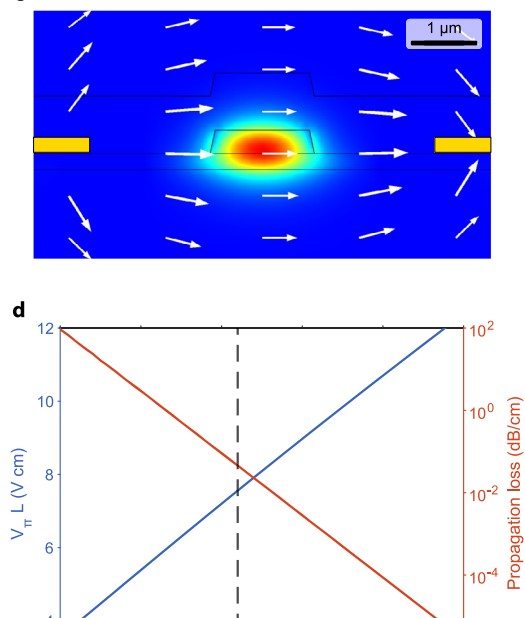

**Fig. 5 | Design of the waveguide and electro-optic modulator. a** Geometry of the nonlinear LN waveguide. The ridge waveguide width is w = 1.2 μm, and its height is h = 300 nm. Slab thickness is s = 200 nm. Waveguides are periodically poled with period Λ = 3.7 μm and covered with SiO$_2$ cladding of thickness c = 700 nm. **b** Effective indices of waveguide modes as a function of wavelength. The fundamental TE mode is highlighted in orange. The dispersion-induced phase velocity

mismatch is compensated through poling with a period proportional to Δn$_{eff}$. **c** Eigenmode solver solution for the fundamental TE mode in the presence of a static electric field. White arrows indicate the direction of the applied bias field. **d** Half-wave voltage-length product and metal-induced propagation loss as a function of the waveguide-electrode gap. The dashed line corresponds to the geometry described in the main text with a waveguide-electrode gap of 1.6 μm.

The bottom layer of electrodes and waveguides are covered with SiO$_2$ deposited with a high-density plasma process (Fig. 4f)[49]. To provide electrical contact to the buried electrodes, we pattern vias by standard SiO$_2$ etch with fluorine chemistry and photolithography (Fig. 4g). Finally, we pattern a top metal layer with the electron beam. This layer provides access to the buried electrodes with an external probe and consists of 200 nm of gold and a 10 nm chromium adhesion layer (Fig. 4h).

In the final step, we prepare the chip facets for light coupling by stealth dicing with a DISCO DFL7340 laser saw[17]. High-energy optical pulses are focused on the substrate to create an array of damage sites. They act as nucleation sites for crack propagation, resulting in a uniform and smooth cleave.

## Waveguide design

We design the waveguide geometry to maximize the normalized nonlinear efficiency for the SHG pumped at 1540 nm and OPA pumped at 770 nm. We find a geometry that results in the normalized efficiency of around $\eta \approx 4000\%/(W\,cm^2)$. We show the waveguide schematic in Fig. 5a, the ridge width is 1.2 μm, height is 300 nm, and slab thickness is 200 nm. We clad the waveguide with around 700 nm of silicon dioxide. For the chosen geometry, we calculate the phase velocity mismatch between 770 and 1540 nm light and compensate for it with periodic poling. Figure 5b shows the effective index of all the supported waveguide modes as a function of wavelength. The difference between the effective index at FH and SH defines poling period $\Lambda = \lambda_{SH}/\Delta n$.

## Electro-optic simulation

We design the geometry of our electro-optic devices to provide phase-tuning functionality without introducing excess loss to the quantum state of light. We model our system using a finite-element mode solver

(COMSOL). We first define the electric field-dependent refractive index[23] as:

$$n'_o = n_o - \frac{1}{2}r_{13}n_o^3 E_z \tag{2}$$

$$n'_e = n_e - \frac{1}{2}r_{33}n_e^3 E_z. \tag{3}$$

The elements of the electro-optic tensor $r_{33} = 31$ pm/V and $r_{13} = 10$ pm/V modify ordinary and extraordinary indices of the crystal as a result of the static field. We solve for the static electric field and follow with eigenmode analysis in the system with a modified refractive index, as shown in Fig. 5c.

We find the half-wave-voltage-length product from the relationship between the effective index and applied voltage $V_\pi L = \lambda/(\partial n_{eff}/\partial V)$. We plot the expected values along with the propagation loss from metal proximity in Fig. 5d. In the device described in the main text, we use a waveguide-electrode gap of 1.6 μm and an electrode length of 2.5 mm. We expect the half-wave voltage in this geometry to be around 30 V, which agrees with the value measured in the experiment. This gap size allows us to keep the propagation loss induced by the metal below 0.05 dB/cm, which is negligible compared to the measured loss resulting from fabrication imperfections around 0.7 dB/cm.

## Experimental setup

We characterize fabricated PICs in two different setup configurations, one for BHD and one for SHG characterization. In both cases, the chip temperature is controlled with a thermistor (Thorlabs TH10K) and thermo-electric cooler (Thorlabs TECF2S, MTD415TE, MTDEVAL1). We

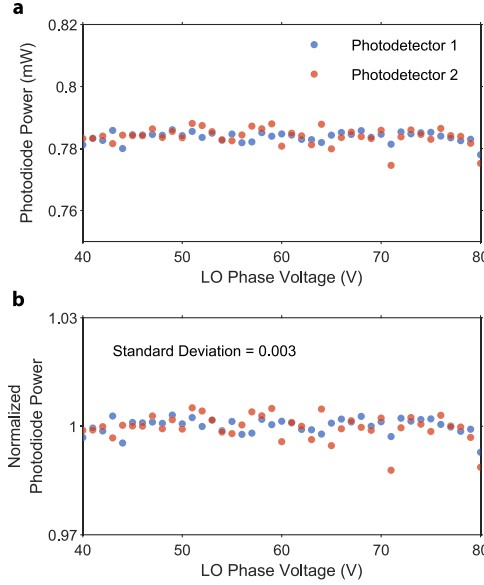

**a**

**b**

**Fig. 6 | Balanced homodyne detection stability. a** Absolute power measured on two photodetectors in the balanced homodyne setup during a single LO phase sweep squeezing measurement. **b** The same data is normalized to the average optical power per detector. The standard deviation of the normalized power is below 0.3%.

operate our device at 45 °C to maximize SHG intensity at the low noise laser operating wavelength.

The setup for BHD measurements is shown in Fig. 1a. For squeezing and quantum-enhanced measurements, we use an ultra-low noise, integrated DFB laser with an emission wavelength of around 1544 nm (Thorlabs ULN15PC). The laser is followed by a high extinction ratio fiber isolator (Thorlabs IOT-G-1550A) and a fiber polarization controller. We introduce an additional variable optical attenuator (HP 8156A) for the shot noise measurement to control the laser power. Then, we couple light to the chip with a lensed SMF-28 fiber (OZ Optics). We collect light into a balanced photoreceiver (Thorlabs PDB425C with RF output conversion gain $1.25 \times 10^5$ V/W) using multi-mode fibers lensed (in-house) and mounted on a v-groove chip with a pitch matching the PIC output waveguide separation. We collect RF spectra with an electronic spectrum analyzer (Rohde & Schwarz FSW26) with a resolution bandwidth set to 60 kHz. Tap ports for FH and SH light filtered within the squeezer subsystem are collected into another lensed multimode fiber, separated by a combination of a 50/50 splitter and optical edge pass filters, and monitored with photodiodes: Newport 1623 Nanosecond Photodetector for FH and Thorlabs APD440A for SH. Electro-optic components on the chip are controlled with DC voltage supplied with three TTi PLH250-P sources. LO phase modulator and the signal interferometer receive RF signal from an arbitrary waveform generator (Rigol DG4102). We mix RF signals and DC bias with bias-tees (ZFBT-4R2G+) and connect to the chip with a triple ground-signal-ground (GSG) probe. We estimate the total insertion loss of RF input at 1.22 dB by combining the loss of two SMA cables (FLC-4FT-SMSM+), DC block (Inmet 8039), and the bias-tee.

We evaluate the system's stability during balanced homodyne measurements based on the DC readout of the individual photodiodes of the balanced detector. We convert the readouts to optical power and plot it for the squeezing measurement dataset in Fig. 6a. Figure 6b shows the same dataset normalized to the mean power of each channel in the BHD. The standard deviation of the relative power is below 0.3%, which allows us to detect small changes to the noise power on the order of a few percent. We also use the DC readouts from the

homodyne detector to normalize the data plotted in the main part of the article.

For the second harmonic characterization, we reconfigure the setup to work with a continuously tunable c-band laser (Santec TSL-550, 1480–1630 nm) passing through a variable optical attenuator (HP 8156A). Next, we tap off 5% of the light for power calibration using a power meter (Newport 918D-IR-OD3R) and pass the light through a polarization controller. Light couples to the chip through an SMF-28 lensed fiber and is collected by the same kind of fiber at the output facet. Then, we out-couple light into free space and split with a 1000 nm short-pass dichroic mirror (Thorlabs DMSP1000) and filter to avoid cross-talk. Finally, we measure light intensity with avalanche photodiodes (Thorlabs APD410A and Thorlabs APD410). We use variable optical attenuators before APDs to prevent saturation and increase the dynamic range of our measurement (HP 8156A and Thorlabs FW102C).

**Squeezer performance**

The integrated squeezer proposed in the main text takes advantage of the second-order nonlinearity in LN waveguides. In this system, both SHG and OPA can be described with coupled-mode equations for the degenerate three-wave mixing process[23]:

$$\frac{\partial}{\partial z}A(z) = -i\sqrt{\eta}\,B(z)A^*(z)\,e^{-i\Delta kz}, \tag{4}$$

$$\frac{\partial}{\partial z}B(z) = -i\sqrt{\eta}\,A^2(z)\,e^{i\Delta kz}. \tag{5}$$

where $A(z)$ and $B(z)$ are amplitudes of the FH and SH modes with frequency relationship $2\omega_A = \omega_B$. $\eta$ corresponds to normalized efficiency, $\Delta k = 2k_A - k_B$ is the phase mismatch between FH and SH modes, which we compensate for with periodic poling. We solve these equations to obtain the magnitude of the power normalized amplitude of the generated SHG, assuming perfect phasematching:

$$|B(z)| = |A(0)|\tanh\left(\sqrt{\eta}\,|A(0)|z\right). \tag{6}$$

In the squeezer subsystem, FH light is filtered out after pumping the SHG, and the SH stays in the waveguide to generate squeezing through OPA. To find the amount of squeezing generated, we will first define field quadratures for the FH mode as follows:

$$X = \frac{A + A^*}{2}, \tag{7}$$

$$Y = i\frac{A - A^*}{2}. \tag{8}$$

Next, we can use equation (4) to find the evolution of the field quadratures X and Y along the waveguide length as follows:

$$\begin{aligned}\partial_z\left(A(z) + A^*(z)\right) &= -i\sqrt{\eta}\left(B(z)A(z)^* - B(z)^*A(z)\right) \\ &= \sqrt{\eta}|B(z)|\left(A(z) + A(z)^*\right).\end{aligned} \tag{9}$$

In the last line, we have chosen to set the phase of $B(z)$ to $-\pi/2$. Performing a similar calculation for the time evolution of the Y quadrature, we arrive at:

$$\partial_z X = \sqrt{\eta}|B(z)|X, \tag{10}$$

$$\partial_z Y = -\sqrt{\eta}|B(z)|Y. \tag{11}$$

Solving these equations yields:

$$X = X(0) \exp(\sqrt{\eta}|B(z)|z), \tag{12}$$

$$Y = Y(0) \exp(-\sqrt{\eta}|B(z)|z). \tag{13}$$

It is clear that the Y quadrature is deamplified while the X quadrature is amplified. This amplification and deamplification process, when applied to shot-noise-limited light, leads to anti-squeezing and squeezing of the quadrature fluctuations. In other words, the variance for quadrature operators under the action of amplification/deamplification is given by:

$$\langle \delta\hat{X}^2 \rangle = \exp(2\sqrt{\eta}|B(z)|z)\langle \delta\hat{X}_0^2 \rangle, \tag{14}$$

$$\langle \delta\hat{Y}^2 \rangle = \exp(-2\sqrt{\eta}|B(z)|z)\langle \delta\hat{Y}_0^2 \rangle, \tag{15}$$

where $\langle \delta\hat{X}_0^2 \rangle$ and $\langle \delta\hat{Y}_0^2 \rangle$ are variances of the quantum state entering the OPA. In the proposed PPLN squeezer subsystem of length $L$, combining equations (6) and (15) defines squeezing as:

$$\frac{\langle \delta\hat{Y}^2 \rangle}{\langle \delta\hat{Y}_0^2 \rangle} = \exp\left(-2L\sqrt{\eta P_{in}}\tanh\left(L\sqrt{\eta P_{in}}\right)\right). \tag{16}$$

**Coherent leakage into the squeezer and LO phase calibration**

In our system, we observe non-perfect filtering of the squeezer sub-section pump at the fundamental frequency. This results from the limited extinction ratio of the filter used on the chip to separate FH from SH after the SHG section. This section explains how we quantify this leakage and use it to calibrate the local oscillator phase.

Our model consists of a tunable beam splitter with two inputs (LO and squeezer output) and two outputs (BHD ports). We calculate a result of a phase modulation imposed on the LO arm on the BHD measurement result, depending on the leakage level. The input state of the system is:

$$\begin{bmatrix} A_{in}^{(1)} \\ A_{in}^{(2)} \end{bmatrix} = \begin{bmatrix} \sqrt{\varepsilon}\alpha_{LO}(t) + \hat{a} \\ \alpha_{LO}(t)e^{i\phi_{LO}}e^{i\phi_M(t)} + \hat{v} \end{bmatrix}. \tag{17}$$

Here, $\varepsilon$ is the power ratio of the coherent leakage to the LO, $\phi_{LO}$ is the LO phase with respect to the signal in the squeezer path, $\phi_M$ is the RF phase modulation, $\hat{a}$ and $\hat{v}$ represent quantum fluctuations in the squeezer and LO paths, respectively. We solve the beam splitter matrix equation:

$$\begin{bmatrix} A_{out}^{(1)} \\ A_{out}^{(2)} \end{bmatrix} = \begin{bmatrix} \sin(\phi_2/2) & \cos(\phi_2/2) \\ \cos(\phi_2/2) & -\sin(\phi_2/2) \end{bmatrix} \begin{bmatrix} A_{in}^{(1)} \\ A_{in}^{(2)} \end{bmatrix} \tag{18}$$

to find the BHD signal generated. This model is equivalent to the signal interferometer described in the main text. For no power leakage, it is biased such that the phase is around $\phi_2 = \pi/2$ but $\varepsilon \neq 0$ results in an imbalance that we have to compensate for.

We solve for the output port amplitudes and use them to find the BHD differential signal defined as $P_{BH} = |A_{out}^{(1)}|^2 - |A_{out}^{(2)}|^2$. The result is

given by:

$$\begin{aligned} P_{BH} = &\cos(\phi_2)|\alpha_{LO}|^2(1-\varepsilon) + 2\sin(\phi_2)\sqrt{\varepsilon}|\alpha_{LO}|^2\cos(\phi_{LO}) \\ &+ \cos(\phi_2)[\alpha_{LO}e^{i\phi_{LO}}\hat{v}^\dagger - \sqrt{\varepsilon}\alpha_{LO}\hat{a}^\dagger + h.c] \\ &+ \sin(\phi_2)[\alpha_{LO}e^{i\phi_{LO}}\hat{a}^\dagger + \sqrt{\varepsilon}\alpha_{LO}\hat{v}^\dagger + h.c] \end{aligned} \tag{19}$$

We separate equation (19) into DC and AC parts. The former defines the locking condition:

$$\cos(\phi_2)(1-\varepsilon) + 2\sin(\phi_2)\sqrt{\varepsilon}\cos(\phi_{LO}) = 0, \tag{20}$$

which is solvable analytically:

$$\phi_2 = \pm\arccos\left[\pm\frac{2\sqrt{\varepsilon}\cos(\phi_{LO})}{\sqrt{4\varepsilon\cos^2(\phi_{LO}) + \varepsilon^2 - 2\varepsilon + 1}}\right]. \tag{21}$$

In addition to locking, we use the DC term to calibrate the LO phase and directly measure the leakage $\varepsilon$. For this purpose, we introduce a periodic modulation to the LO phase shift, such that $\phi_{LO} = \phi_{LO} + \phi_M(t) = \phi_{LO} + \pi V_{p-p}/(2V_\pi)\sin(\Omega t)$. $\Omega$ is the modulation frequency (here set to 72 MHz), $V_\pi$ is the half-wave voltage, and $V_{p-p}$ is the peak-to-peak modulation voltage. We focus on the DC part of the equation (19), apply a standard Taylor series expansion to the phase modulation term, and write the resulting signal in the frequency domain at the peak frequency of the modulation:

$$P_{RF}(\Omega) = \frac{2\varepsilon R^2 P_{LO}^2}{Z}\left(\frac{\pi}{2}\right)^3\frac{V_{p-p}}{V_\pi}\sin^2\left(\frac{V_{DC}}{V_\pi}\pi\right), \tag{22}$$

$V_{DC}$ is the DC voltage bias at the LO phase shifter. Signal strength depends on the local oscillator power $P_{LO}$, detection responsivity $R$, the impedance of the detector $Z$, and the leakage $\varepsilon$. We fit equation (22) in the main text to find $\varepsilon$ and $V_\pi$.

**Impact of leakage on the squeezing visibility**

The AC part of the equation (19) probes both the noise of the prepared quantum state and the noise of the LO. It results in the noise measured in the main text:

$$\begin{aligned} P_{BH}^{(AC)} = &\cos(\phi_2)[\alpha_{LO}e^{i\phi_{LO}}\hat{v}^\dagger - \sqrt{\varepsilon}\alpha_{LO}\hat{a}^\dagger + h.c] \\ &+ \sin(\phi_2)[\alpha_{LO}e^{i\phi_{LO}}\hat{a}^\dagger + \sqrt{\varepsilon}\alpha_{LO}\hat{v}^\dagger + h.c]. \end{aligned} \tag{23}$$

Note that it reduces to the usual BHD expression for $\varepsilon = 0$ and $\phi_2 = \pi/2$. Our experiment introduces two modifications to the classical BHD picture. One is the leakage $\varepsilon \neq 0$, which results in probing the local oscillator noise $\hat{v}$, in addition to measuring the squeezed state $\hat{a}$. The other modification is varying the splitting ratio of the output beam splitter $\phi_2 \neq \pi/2$. This adds a term proportional to $\cos(\phi_2)$, resulting in changing the measured noise characteristics. As a result, measured squeezing can be reduced. This section explains how the power leakage $\varepsilon$ impacts observable squeezing. We first rewrite the equation (23) with the quadrature operators for the squeezing and LO paths $\hat{a} = \delta\hat{X} - i\delta\hat{Y}$ and $\hat{v} = \delta\hat{X}_{LO} - i\delta\hat{Y}_{LO}$:

$$\begin{aligned} P_{DIFF}^{(AC)} = &2\sin(\phi_2)|\alpha_{LO}| \\ &\times\left[\cos(\phi_{LO})\delta\hat{X} - \sin(\phi_{LO})\delta\hat{Y} + 2\sqrt{\varepsilon}\delta\hat{X}_{LO}\cdot\right] \\ &+ 2\cos(\phi_2)|\alpha_{LO}| \\ &\times\left[\cos(\phi_{LO})\delta\hat{X}_{LO} - \sin(\phi_{LO})\delta\hat{Y}_{LO} - 2\sqrt{\varepsilon}\delta\hat{X}\right] \end{aligned} \tag{24}$$

This is an expected result for the BHD applied toward phase-sensitive probing of the quadratures of a squeezed state with an additional term that probes the noise of the LO. We assume the LO noise to be phase-

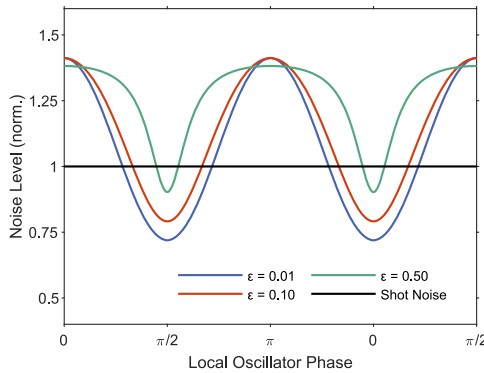

**Fig. 7 | Effect of coherent leakage on squeezing visibility.** We calculate the expected noise assuming 1.5 dB squeezing and anti-squeezing and coherent leakage in the squeezer subsystem. The noise values correspond to the values on the chip (excluding fiber-to-chip coupling efficiency) and are normalized to the shot noise. We plot the noise as a function of the local oscillator phase for three different values of coherent leakage $\varepsilon = 0.01$, 0.1, and 0.5 ($\sqrt{\varepsilon} = \alpha_{\text{Leakage}}/\alpha_{\text{LO}}$).

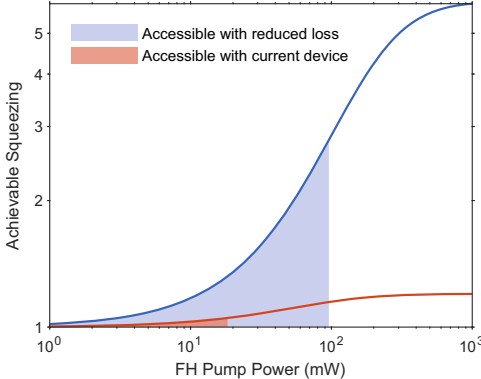

**Fig. 8 | Calculated squeezing with reduced loss.** We calculate the expected squeezing for the proposed device with reduced detection and laser coupling loss. We assume a total laser coupling loss of 0.6 dB and a total detection loss of 0.8 dB. The blue line corresponds to the expected squeezing after reducing optical loss in the system. The orange line corresponds to the squeezing achievable with the current system, with 20% detection efficiency. The shaded areas are accessible with the current integrated DFB laser for both cases.

insensitive and shot-noise-limited. The signal variance is proportional to the power measured by the spectrum analyzer and given by equation (25). We use this result as a model for our squeezing measurement in the main text.

$$
\begin{aligned}
\left\langle \delta P_{\text{DIFF}}^{(AC)2} \right\rangle = 4\,|\alpha_{\text{LO}}|^2 \Big[ &\cos^2(\phi_2)\Big(\cos^2(\phi_{\text{LO}})\left\langle \delta \hat{X}_{\text{LO}}^2 \right\rangle + \sin^2(\phi_{\text{LO}})\left\langle \delta \hat{Y}_{\text{LO}}^2 \right\rangle + 4\,\varepsilon\left\langle \delta \hat{X}^2 \right\rangle \Big) \\
&+ \sin^2(\phi_2)\Big(\cos^2(\phi_{\text{LO}})\langle \delta \hat{X}^2 \rangle + \sin^2(\phi_{\text{LO}})\langle \delta \hat{Y}^2 \rangle + 4\varepsilon\left\langle \delta \hat{X}_{\text{LO}}^2 \right\rangle \Big) \\
&+ 2\,\sin(2\phi_2)\,\sqrt{\varepsilon}\,\cos(\phi_{\text{LO}})\Big(\left\langle \delta \hat{X}_{\text{LO}}^2 \right\rangle - \left\langle \delta \hat{X}^2 \right\rangle \Big) \Big]
\end{aligned}
$$
(25)

We illustrate the effects of coherent leakage on squeezing visibility in Fig. 7. We plot the normalized BHD signal using the formula (25) as a function of the local oscillator phase for 1.5 dB of squeezing and anti-squeezing. We assume that the system is locked such that the homodyne detector is balanced, following formula (21) and plot all the values normalized to the shot noise. We plot the results for three squeezer leakage ratios: $\varepsilon = 0.01$, 0.1, and 0.5. The leakage on the order of less than 10% does not change the squeezing visibility significantly. However, excessive leaks can have a more severe impact. Note that we do not include the fiber-to-chip coupling efficiency in this plot to isolate the leakage effect from the detection loss.

### Effects of loss in the system
The total detection chain efficiency is a major challenge in characterizing squeezed light. This is especially consequential when working with complex PICs like the one described in the main text. In our experiment, the total detection chain efficiency was around $\zeta \approx 20\%$, this leads to the reduction of squeezing visibility:

$$
\left[ \frac{\left\langle \delta \hat{Y}^2 \right\rangle}{\left\langle \delta \hat{Y}_{\text{vac}}^2 \right\rangle} \right] = \left( \zeta \left[ \frac{\left\langle \delta \hat{Y}^2 \right\rangle}{\left\langle \delta \hat{Y}_{\text{vac}}^2 \right\rangle} \right]_{\text{On-Chip}} + (1 - \zeta) \right).
$$
(26)

## Table 1 | Summary of the sources of loss in the detection chain

| Loss source | Loss (dB) | Determination |
|---|---|---|
| Waveguide propagation | 0.80 | Q-factor measurement in a ring resonator on the same chip |
| Lensed multimode fiber collection | 5.40 | Coupling test to a straight waveguide |
| Detector quantum efficiency | 0.80 | Photodetector documentation |

We summarize all the sources of loss that impact reported total detection efficiency in Table 1. In our measurement, the main limitation is a result of the loss at the interface between the chip facet and collection fibers.

Reducing the overall losses in the system is a straightforward way to improve the device's performance. Reducing the fiber-to-chip coupling loss down to 0.6 dB has been established by engineering the waveguide termination[40–42]. Moreover, the propagation loss can be reduced to 0.2 dB by reducing the waveguide length where the squeezed state can experience loss. The quantum efficiency of the detector used in this study can be improved to 99.5% using commercially available photodiodes. We use these values to estimate potentially achievable squeezing in the proposed device. In this case, the total detection loss is 0.8 dB, and the total insertion loss of the laser is 0.6 dB. Using these values, we plot the expected achievable squeezing with respect to the laser power in Fig. 8. Assuming no squeezer leakage, we compare the calculated values in a system with reduced loss (blue) to the current system with a total detection efficiency of around 20% (orange). We estimate that the same integrated DFB laser, as we used in the main text, would facilitate access to the blue-shaded region and result in the maximum squeezing factor of 2.8 at the highest achievable on-chip power of 96 mW.

### Fundamental harmonic filter design and characterization
We designed the FH filter based on a directional waveguide coupler. We calculate the coupling ratio by simulating a cross-section of two adjacent waveguides in a finite-element mode solver (COMSOL). We extract effective indices for the symmetric and anti-symmetric modes, their difference defines the coupling strength. Then, we find the length of the coupler necessary to transfer 100%. We choose the gap between waveguides to avoid SH light coupling. Our design length is 380 μm, and the gap between waveguides is 1 μm.

Due to the fabrication imperfections and bend contributions to the coupling, the actual devices differ from the ideal design. We verify the filter performance experimentally by including it inside a racetrack resonator with a total length of around 2 mm, as shown in the inset of Fig. 9a. The filter contributes to the intrinsic loss of the cavity and has a clear wavelength dependence. We measure it by sweeping a tuneable laser coupled to a bus waveguide and fitting lorentzians to the cavity modes. Figure 9a shows the measured intrinsic quality factors. Assuming that the propagation loss of the waveguide is constant across our measurement bandwidth of 1480–1620 nm, we extract the

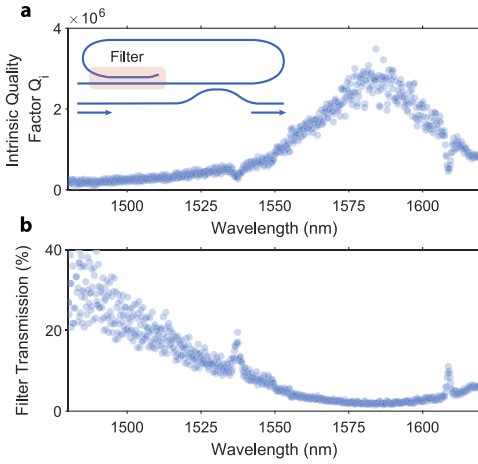

**Fig. 9 | Fundamental harmonic filter characterization. a** Measured intrinsic quality factors of a snail resonator as a function of wavelength. The inset shows resonator design—the characterized filter is a part of the cavity and contributes to the intrinsic loss. **b** Extracted filter transmission based on the cavity measurements. The filter's transmission at 1544 nm is around $(8 \pm 1)$%. Increased loss at around 1535 and 1610 nm results from cavity mode crossing and is unrelated to the filter.

loss contribution from the coupler and plot it in Fig. 9b. The fabricated devices perform best at around 1585 nm and transmit around $(8 \pm 1)$% of light at the squeezer subsystem operation wavelength of 1544 nm. We use three back-to-back filters in the squeezer subsection and observe a suppression of around 18 dB. The excess leakage can result from fabrication imperfections and stray light coupling into the system. The increased density of waveguides in the squeezing PIC can result in more severe electron beam proximity effects and changes in the coupler dimensions. Stray light can propagate in the slab or insulator layer and be parasitically coupled to the circuit.

## Data availability
The data sets generated during and/or analyzed during this study are available from the corresponding author on request.

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

## Acknowledgements

This work was supported in part by the Defense Advanced Research Projects Agency (DARPA) LUMOS program (Grant No. HR0011-20-2-0046, received by A.H.S.-N.) and the DARPA Young Faculty Award (YFA, Grant No. D19AP00040, received by A.H.S.-N.). We also thank NTT Research for their financial and technical support, and the US Department of Energy for their support through Grant No. DE-AC02-76SF00515, received by A.H.S.-N. In addition, the US Air Force Office of Scientific Research provided a MURI grant (Grant No. FA9550-17-1-0002, received by A.H.S.-N.) that supported this research. H.S.S. acknowledges support from the Urbanek Family Fellowship, and V.A. was partially supported by the Stanford Q-Farm Bloch Fellowship Program. This work was also performed at the Stanford Nano Shared Facilities (SNSF), supported by the National Science Foundation under award ECCS-2026822. We also acknowledge the Q-NEXT DOE NQI Center and the David and Lucille Packard Fellowship for their support. D.D. acknowledges support from the NSF GRFP (No. DGE-1656518). H.S.S. and V.A. thank Kevin Multani, Christopher Sarabalis, and Michael Stefszky for discussions and technical support.

## Author contributions

H.S.S. designed the device. H.S.S., T.P., and A.Y.H. fabricated the device. H.S.S., V.A., T.P.M., and O.T.C. developed the fabrication process. M.M.F. and A.H.S.-N. provided experimental and theoretical support. H.S.S., V.A., T.P., and D.J.D. performed the experiments and analyzed the data. H.S.S., V.A., and A.H.S.-N. wrote the manuscript. H.S.S., V.A., T.P.M., and A.H.S.-N. conceived the experiment, and A.H.S.-N. supervised all efforts.

## Competing interests

The authors declare no competing interests.
