## [Peer Review File · Nature Communications]

Integrated Quantum Optical Phase Sensor in Thin Film Lithium NiobateEditorial Note:

This manuscript has been previously reviewed at another journal that is not operating a transparent peer review scheme. This document only contains reviewer comments and rebuttal letters for versions considered at *Nature Communications*.

REVIEWERS' COMMENTS

Reviewer #2 (Remarks to the Author):

The authors considerably improved the manuscript and addressed my comments appropriately. I still have a couple of points that will further improve the manuscript:

- In the manuscript you mention that the LO power stays within 0.3%. When tuning the power from shot noise calibration measurement to squeezing measurement, is the error on the LO power also within these bounds? This is an important systematic error that should be mentioned as well.
- Following that you have 0.2% statistical error on the amount of squeezing, you should mention the systematic error as well by writing 2.7 ± 0.2 (statistical) ± 0.3 (systematic) or similarly.
- At several occasions you mention that the LO power is "about 7.8 mW". Given that you had to characterize the error and the importance of the power to claim the amount of squeezing you measure, it would be better to replace the LO power with the exact value including the error.
- The sentence "Note that the linearity of our measurement implies that the noise is not correlated with the laser power" is misleading as of course the amount of (shot) noise you measure depends on the LO power. Please rephrase.

Reviewer #3 (Remarks to the Author):

The authors have made substantial changes to address the comments of all reviewers, and provided articulate responses to the concerns raised. I am happy to recommend publication in *Nature Communications*.

There are only two areas where I feel their responses did not entirely address my concerns:

- 1) The query I raised about useful applications given inefficiencies in SHG and OPA, and the comparison to performance were these removed and classical sensing performed. In their response, the authors rightly point to practical limitations on much light can be used in lithium niobate circuits. However, these limitations will also apply to the pumps for the SHG and OPA when using squeezed light. The total power of the squeezed light will presumably end up substantially lower than the input pump intensities. Removing the SHG and OPA, would allow the full input intensity to be used in the sensor and therefore, I expect, far better absolute sensitivity in most scenarios.
- 2) I think the authors may have misunderstood my minor query (4). I understand their comment that changing the phase of the local oscillator is equivalent to changing the phase of the squeezing. However, there is a third phase that - as I understand it - is important. That is the phase of the signal modulation that they apply to show squeezing-improved SNR. If this phase doesn't correctly align with the squeezed quadrature of the light, then the measured signal amplitude when the local oscillator is set to measure the squeezed quadrature will not be the maximum signal amplitude. In this case, the optimal SNR will occur for a local oscillator that is different than that of the squeezed quadrature, and it is likely, given the low level of squeezing, not to exhibit any quantum enhancement.

REVIEWER COMMENTS

Reviewer #2 (Remarks to the Author):

The authors considerably improved the manuscript and addressed my comments appropriately. I still have a couple of points that will further improve the manuscript:

- In the manuscript you mention that the LO power stays within 0.3%. When tuning the power from shot noise calibration measurement to squeezing measurement, is the error on the LO power also within these bounds? This is an important systematic error that should be mentioned as well.

In the shot noise calibration, we actively measure LO power and noise simultaneously, hence the uncertainty there only comes from the detector noise, equivalent to about 0.2 uW, or relative deviation of 3×10^{-5} . The uncertainty during squeezing is higher than that because we are actively trying to lock the system to the same LO power as we measured during the shot noise calibration.

The stability of the LO shown in the manuscript (Fig. 6) comes directly from the same dataset as measured squeezing (Fig. 3c). During the entire measurement we actively lock the input and output MZIs on chip and monitor the two outputs.

We add the following clarifications to the manuscript:

“Note that the noise and LO power are recorded simultaneously, and the detector noise determines the uncertainty of the LO power below a relative deviation of 3×10^{-5} .”

*“The relative deviation of the LO power **during the squeezing measurement** is below 0.3%, which allows us to detect changes to the noise on the order of several percent (see Methods for details).”*

- Following that you have 0.2% statistical error on the amount of squeezing, you should mention the systematic error as well by writing 2.7 ± 0.2 (statistical) ± 0.3 (systematic) or similarly.

Thank you for this comment, we added a clarification in the manuscript:

“We measure $(2.7 \pm 0.2$ (stat) ± 0.3 (syst)) % of squeezing and $(12.3 \pm 0.2$ (stat) ± 0.3 (syst)) % of anti-squeezing.”

- At several occasions you mention that the LO power is "about 7.8 mW". Given that you had to characterize the error and the importance of the power to claim the amount of squeezing you measure, it would be better to replace the LO power with the exact value including the error.

We agree, and we added this modification to the main text. When we talk about the LO power during the squeezing measurements, we quote it as 7.84 ± 0.03 mW. When discussing the shot noise calibration and setpoint for the squeezing measurement, we only mention 7.84 mW.

- The sentence "Note that the linearity of our measurement implies that the noise is not correlated with the laser power" is misleading as of course the amount of (shot) noise you measure depends on the LO power. Please rephrase.

We have rephrased it to:

"Note that the linearity of our measurement implies that there is no excess intensity noise from the laser."

Reviewer #3 (Remarks to the Author):

The authors have made substantial changes to address the comments of all reviewers, and provided articulate responses to the concerns raised. I am happy to recommend publication in Nature Communications.

There are only two areas where I feel their responses did not entirely address my concerns:

1) The query I raised about useful applications given inefficiencies in SHG and OPA, and the comparison to performance were these removed and classical sensing performed. In their response, the authors rightly point to practical limitations on much light can be used in lithium niobate circuits. However, these limitations will also apply to the pumps for the SHG and OPA when using squeezed light. The total power of the squeezed light will presumably end up substantially lower than the input pump intensities. Removing the SHG and OPA, would allow the full input intensity to be used in the sensor and therefore, I expect, far better absolute sensitivity in most scenarios.

Firstly, we would like to point out that even if we ignore the issues related to optical damage and detector saturation, a splitting ratio between how much light is sent through the LO vs. the squeezing path can be found to realize a *power-optimal measurement*. Consider some future implementation of our device where 90% of the power is sent through the LO path, while 10% is sent through the SHG/OPA path. Let's assume we succeed in making the detection path sufficiently low-loss that the 10% through the SHG/OPA path leads to 10 dB squeezing. Then our measurement effectively behaves as one where 900% of the light is sent through the LO path. Therefore, given the same amount of total power, a significantly more precise measurement would, in principle, be possible.

In principle, with the demonstrated device, we should have been able to operate in such a regime. However, leakage and losses made operation at much higher power impractical, and so we operated the system in a regime where most of the power was sent through the SHG/OPA path. Therefore our demonstration does not show a *power-optimal measurement*. We leave this to future work, and hopefully, we will be able to realize such a quantum advantage in sensing in the next few years.

On the other hand, there are practical reasons why lower powers in the modulator would be desirable. We were referring to these in our response.

The parasitic nonlinear effects in the interferometer are critical to the operation of the sensor. Especially photorefraction can make the MZM unstable to the point of becoming unpredictable. In addition, parasitic frequency conversion can introduce effective loss. The same effects in the SHG or OPA waveguides are less critical to the operation of our device. They may limit the SHG or OPA efficiency but will not cause the sensor to be unreliable. We put more emphasis on the photorefractive effect in the manuscript:

“Especially the photorefractive effect can cause instability of electro-optic modulators.”

The detector saturation and nonlinearities are other factors that limit practically the power one can use in a sensor. This is already mentioned in the manuscript. We agree that if there was no limitation to the amount of power in the interferometer, increasing it is a straightforward way to improve SNR. We already mentioned this in the manuscript:

“To increase the SNR within the confines of the QNL, we must increase N -- by using more optical power or extending the measurement time. These approaches are often undesirable or impossible due to technical or fundamental limitations.”

2) I think the authors may have misunderstood my minor query (4). I understand their comment that changing the phase of the local oscillator is equivalent to changing the phase of the squeezing. However, there is a third phase that - as I understand it - is important. That is the phase of the signal modulation that they apply to show squeezing-improved SNR. If this phase doesn't correctly align with the squeezed quadrature of the light, then the measured signal amplitude when the local oscillator is set to measure the squeezed quadrature will not be the maximum signal amplitude. In this case, the optimal SNR will occur for a local oscillator that is different than that of the squeezed quadrature, and it is likely, given the low level of squeezing, not to exhibit any quantum enhancement.

Thank you for clarifying your point. Our proposed measurement is not sensitive to the RF signal phase. It results in an additional shift φ_{Sig} in the signal interferometer phase $\phi_2(t) = \phi_{2,0} + \pi V_{pp}/(2V_{\pi}) \sin(\Omega_{Sig} t + \varphi_{Sig})$. The measured RF peak power can be found by inserting $\phi_2(t)$ to eq. 19, performing Fourier transform, including the photodetector's responsivity, and taking the absolute value squared. The result does not depend on φ_{Sig} . Another way of seeing this is that RF source and laser do not need to be phase coherent.